# Detection of Lymphoid Markers (CD3 and PAX5) for Immunophenotyping in Dogs and Cats: Comparison of Stained Cytology Slides and Matched Cell Blocks

**DOI:** 10.3390/vetsci10020157

**Published:** 2023-02-15

**Authors:** Filipe Sampaio, Carla Marrinhas, Luísa Fonte Oliveira, Fernanda Malhão, Célia Lopes, Hugo Gregório, Carla Correia-Gomes, Ricardo Marcos, Mario Caniatti, Marta Santos

**Affiliations:** 1Cytology and Hematology Diagnostic Services, Laboratory of Histology and Embryology, Department of Microscopy, ICBAS–School of Medicine and Biomedical Sciences, University of Porto (U.Porto), Rua de Jorge Viterbo Ferreira, 228, 4050-313 Porto, Portugal; 2Laboratório INNO, 4710-503 Braga, Portugal; 3Hospital do Baixo Vouga, OneVet Group, 3750-742 Águeda, Portugal; 4Anicura CHV Porto, 4100-320 Porto, Portugal; 5Animal Health Ireland, Carrick on Shannon Co., N41 WN27 Leitrim, Ireland; 6Dipartimento di Medicina Veterinaria e Scienze Animali (DIVAS), Università degli Studi di Milano, 26900 Lodi, Italy; 7Oncology Research, UMIB—Unit for Multidisciplinary Research in Biomedicine, ICBAS, University of Porto, 4050-313 Porto, Portugal; 8ITR—Laboratory for Integrative and Translational Research in Population Health, 4050-600 Porto, Portugal

**Keywords:** cytology, lymphoma, dog, cat, phenotype, immunocytochemistry, immunohistochemistry, cell blocks

## Abstract

**Simple Summary:**

The B- or T-cell phenotype of a canine or feline lymphoma is relevant for treatment selection and prognosis. This phenotype is determined by complementary techniques and there is some evidence that the previously stained cytology (RSC) slides used for routine diagnose of lymphoma are suitable for assessing B and T-cell phenotype by immunocytochemistry. Still, immunodetection of the lymphoid markers can also be achieved with cell blocks (CB) prepared from effusions fluids or needle rinses. A comparison between these two RSC and CB is missing in the literature. In this study, lymphoid markers (CD3 and PAX5) were simultaneously studied in RSC and matched CB from 53 lymphomas and 4 chylous (lymphocyte-rich) effusions from dogs and cats. The influence of pre-analytical variables (species, time of archive, type of specimen, and coverslipping) and the interobserver agreement were assessed. Fewer CD3+ lymphocytes were identified in RSC, but the PAX5 positivity in RSC and CB had a substantial agreement. Immunodetection of CD3 and diagnosing a T-cell population on RSC were more difficult. Immunophenotyping was inconclusive in 54% RPSC and in 19% CB. The interobserver reproducibility of immunophenotyping on CB was substantial and higher than in RSC. The performance of RSC from effusions and feline samples was unsatisfactory. The detection of lymphoid markers on RSC was affected by pre-analytical variables, whilst CB was more consistent for assigning a lymphoma phenotype.

**Abstract:**

Immunolabeling on Romanowsky-stained cytology (RSC) slides can be used, although there is limited evidence of its suitability for phenotyping canine and feline lymphomas. A comparison with matched cell blocks (CB) is missing. Immunolabeling on RSC and CB was compared for lymphoid markers (CD3 and PAX5) in 53 lymphomas and 4 chylous effusions from dogs and cats. The influence of pre-analytical variables (species, time of archive, type of specimens and coverslipping) and the interobserver agreement among the 2 observers was assessed. Fewer CD3+ lymphocytes were identified in RSC, while the PAX5 positivity by RSC and CB had a substantial agreement. Immunodetection of CD3 and the diagnosis of a T-cell population on RSC was more difficult. Lower intensity and higher background were noted in RSC. Immunophenotyping was inconclusive in 54% RSC and 19% CB. The interobserver reproducibility of immunophenotyping on CB was substantial, being higher than in RSC. The immunolabeling performance on the RSC of effusion and feline samples was unsatisfactory. The detection of lymphoid markers, especially membranous antigens in retrospective RSC, is affected by the pre-analytical variables: species, time of the archive, and type of specimens. CB are a more consistent type of sample for immunophenotyping purposes.

## 1. Introduction

Lymphoma is a malignant tumor of lymphoid cells that proliferate within solid organs, whether in primary lymphoid structures, such as bone marrow or thymus or in secondary ones, such as lymph nodes, spleen, and mucosa-associated lymphoid tissue in the gastrointestinal tract. Lymphomas are one of dogs and cats’ most recognized and described neoplasia [1,2,3,4].

Cytological examination of fine-needle aspiration (FNA) samples or cavitary effusion fluids is an inexpensive, first-line and reliable method for diagnosing most of the lymphoma cases in dogs and cats [3,4]. The complete diagnostic work-up of a lymphoma case includes the clinical presentation, cell morphology features and grading, immunophenotype, and molecular biology analysis [5]. Nowadays, phenotyping techniques comprise cell block (CB) immunolabeling, immunohistochemistry of tissues (biopsies), flow cytometry, and PCR for Antigen Receptor Rearrangements (PARR). These are the main ancillary techniques used for canine and, to a lesser extent, for feline lymphoma. Assessing the phenotype and clonality of lymphoproliferative diseases is essential to confirm or exclude the diagnosis when the morphological evaluation raises doubts (e.g., intestinal small cell lymphoma in cats) [6]. The immunophenotype of lymphoid cells is determined with specific monoclonal and/or polyclonal antibodies. Specifically, the membranous CD3 is used as a T-cell marker, whereas CD79α, CD21, or CD20 (all membranous antigens) and, more recently, the nuclear PAX5 can be used as B-cell markers [7,8,9,10]. This latter B-cell marker was critical in diagnosing T-cell lymphomas presenting CD79α and CD20 positive cells, because PAX5 was invariably negative [11,12]. The B- or T-cell phenotype, as well as the degree of maturation of neoplastic lymphoid cells, has proven to be useful for the classification and prognosis of canine and feline lymphomas [2,7,13]. The gold standard test for assessing the phenotype of lymphoma cells in dogs and cats is immunohistochemistry applied to biopsy tissues. However, cytology is often the first line of diagnostic tests used in most canine and feline lymphomas. Moreover, in many veterinary settings, cytology is the only affordable test for many owners [14]. It has been reported that 70% to 90% of canine lymphomas are diagnosed by cytology [15,16]. Hence, maximizing the information obtained from such specimens is desirable and valuable for the prompt clinical management of lymphoma in dogs and cats.

The first report describing the use of cytological specimens for determining the phenotype of lymphoma in dogs was published more than 30 years ago using cytospin smears obtained from FNA needle rinses of canine lymph nodes [17]. In this pioneering study, cytology slides were kept unstained until immunolabeling. However, the number of lymphoid cells and their preservation is unknown by using unstained slides prior to immunolabeling. To overcome these limitations, two recent studies showed that lymphoid markers could be detected in previous Romanowsky-stained cytology (RSC) slides [10,18]. Currently, the suitability of RSC for confirming or immunophenotyping feline lymphomas is largely unknown. Moreover, it has never been entirely determined if archived cytology specimens can be used for retrospective studies of canine and feline lymphoma. Every immunolabeling reaction can be influenced by several factors occurring outside the laboratory and before the immunolabeling itself. These are included in the so-called pre-analytical variables. Such variables differ, whether the immunolabeling (i.e., the test, the analytical phase) is performed in cells or tissues. For RSC, the pre-analytical variables include: the animal species, the type of specimen, the previous stain used and the exposure to air (i.e., whether the slides were coverslipped). To better evaluate the influence of pre-analytical variables, the immunostaining should be compared in the same set of cells (i.e., effusion cells on RSC compared to those on CB, obtained from the same effusion), instead of comparing different populations (e.g., immunolabeling of effusion cells on RSC compared to immunohistochemistry of lymph node/mass biopsies).

In human pathology, CB prepared from surplus effusion fluids or needle rinses are nowadays considered the most suitable specimens for prospective or retrospective studies [19]. In canine and feline lymphomas, few studies described the utility of CB, prepared according to different methods [8,9,20]. In those reports, the success of immunophenotyping canine T-cell lymphomas varied, from a low percentage using liquid-based cytology and commercial kits in lymph node samples [9] to a high success rate using the cell tube block technique in effusion samples [20].

In veterinary medicine, it is unknown if immunolabeling on CB has advantages over RSC. Elucidating this issue would have a major clinical relevance, since clinicians and oncologists may decide to obtain cytology samples or cytology plus a CB (from needle rinses or effusion fluids) in a canine or feline case suspected of a lymphoma. This would be particularly relevant when the owners are unwilling or cannot afford to submit the animal to a more invasive and expensive procedure, such as obtaining a surgical biopsy specimen for histopathology [8]. Considering this, in the present study, we aimed to: (1) compare the immunolabeling results in RSC and the matched CB (obtained from surplus effusion fluids or needle rinse fluid) of canine and feline lymphomas using CD3 and PAX5 markers; (2) determine the interobserver agreement in defining the immunophenotype by RSC and by CB; (3) assess the influence of pre-analytical variables, such as species, time of archive and coverslipping on positivity, intensity, background, and non-specific staining in immunolabeling with a membranous and a nuclear lymphoid marker on RSC. All analyses were performed by consensus of two cytopathologists in a multi-headed microscope (considered as observer 1; one with more than 5 years and another with almost 20 years of experience) and by a board-certified pathologist (regarded as observer 2; with more than 30 years of experience and experience in the specific topic of canine and feline lymphomas).

## 2. Material and Methods

### 2.1. Sample Collection and Case Selection

Archived canine and feline lymphocyte-rich effusions and organ lymphoma cases were retrospectively selected from the archives of the Cytology Diagnostic Services, University of Porto, and from the archive of a private veterinary hospital (Hospital do Baixo Vouga, OneVet Group, Águeda, Portugal). Cases were initially selected based on the availability of a representative CB performed at the time of the cytological diagnosis. A total of 57 cases (30 dogs and 27 cats) were enrolled, comprising 26 cavitary effusions, 31 FNA cytology specimens, and the matched CB. As for the CBs from effusions, these were obtained directly from the liquid when a diagnosis of lymphoma was made or when lymphoma could not be completely ruled out, and surplus fluid was available. The CB was obtained from the needle rinses after the FNA of the lymph nodes or organs, for nodal or organ lymphoma cases, respectively. Two cytopathologists (FS and MS) reviewed cytology slides blinded to the final diagnosis (i.e., lymphoma phenotype) and cases were only enrolled if at least one representative slide with good cell preservation was available. Cases were excluded whenever there was low cellularity or poor cell preservation. Furthermore, data regarding the pre-analytical variables of the RCS were recorded; these included: species, time of the archive, type of specimen, and presence of coverslip.

The CB was obtained following the cell tube block technique [20,21]. Briefly, capillary tubes were filled with effusion and needle rinse fluids, sealed with clay, and spun in a microhematocrit centrifuge (12,700 g for 5 min). Poorly cellular effusion samples were first concentrated, and the capillary tube was filled with the sediment. In all non-bloody samples, a high-density solution (Percol, Sigma-Aldrich, St. Louis, MO, USA) was introduced in the capillary tube to ensure the proper separation of the cell pellet and clay. Tubes were cut at the liquid-solid interface, and routinely processed for paraffin embedding. The fixation time and all the paraffin-embedding and section procedures of the CB were standardized in a single laboratory.

### 2.2. Immunolabeling on Romanowsky-Stained Cytology Slides and Immunolabeling of Sections from the Cell Block Protocol

RSC was scanned (Olympus VS110 slide scanner, Olympus, Japan; ×40 objective, which ensures a view up to 100× magnification using digital zoom settings) for archive and reviewing purposes. Coverslip was removed by soaking the slides in xylene for 6 days (independently of the time of archive). On a single cytology slide, two separated staining areas were defined for immunolabeling on RSC (by scrapping with a blade). The matched CB was routinely sectioned (3–4 µm sections), and one section stained with H&E to confirm the cell representativeness. The subsequent sections were placed on precoated slides and later on used for immunolabeling on CB.

The RSC protocol was optimized based on a recently published study [10]. In this protocol validation phase, the fixation with cold acetone and ethanol and antigen retrieval with low pH citrate buffer produced comparable results in both unstained frozen cytology smears and paired RSC [10]. It should be stressed that positive controls obtained from normal canine and feline lymph nodes (collected during necropsies) and negative controls were included in all the immunostaining runs of RSC and CB.

After the coverslip was detached, the cytology slides were immersed in 95% ethanol for 10 min, fixed with cold acetone for 15 min, and left to air dry for ≈5 min [10]. CB and cytology samples were subsequently processed similarly (batches always included the two sets of samples). Antigen retrieval was achieved by slide immersion in citrate buffer (pH 6.0) for 3 min at high pressure, using a pressure cooker. Then, slides were allowed to cool down to room temperature. A polymer-based immunostaining kit (Novolink Max Polymer Detection System, Novocastra, Leica Biosystems Newcastle) was used to develop the staining. Primary antibodies anti-PAX5 (mouse monoclonal antibody Clone MX017 ready-to-use, Master Diagnóstica, Granada, Spain) and anti-CD3 (polyclonal rabbit A0452c clone, Dako, Glostrup, Denmark) at 1:400 dilution in phosphate-buffered saline (PBS) 5% bovine serum albumin (BSA) were applied for 2 h at room temperature in a humidity chamber. PBS replaced the primary antibody with 5% BSA in negative controls. Blocking endogenous peroxidase activity was achieved by 10 min of incubation with 3% hydrogen peroxide in methanol. The chromogen 3,3′-diaminobenzidine (DAB chromogen, Kit Novolink Max Polymer Detection System, Novocastra) was used to visualize the final reaction, and nuclear counterstaining was performed with Mayer’s Hematoxylin.

### 2.3. Immunolabeling Evaluation and Immunophenotyping

Semi-quantitative evaluation of the immunolabeling on RSC and CB results was performed by consensus of two cytopathologists (FS and MS—considered as observer 1) in a multi-headed microscope. As previously described, a 0–4 scoring scheme was applied when evaluating the positive cell percentage for each antibody, as previously described [22]. Briefly, score 0 meant no positive cell for the marker, score 1 ≤ 25%, score 2 > 25 and ≤50%, score 3 > 50 and ≤75%, and score 4 > 75% of positive cells.

The intensity and background (if present) were classified as weak (level 1), moderate (level 2), or intense (level 3) compared to the controls. The presence of non-specific staining (i.e., the pattern of immunolabeling, for example, cytoplasmic in a nuclear marker or staining in non-lymphoid cells) was also recorded. The evaluation of the immunolabeling on CB was blinded to the results of RSC and vice versa. The immunophenotype was classified based on the number and distribution of cells positive for CD3 and PAX5 by RSC [12]. The same approach was used for the immunostained CB slides (being the evaluation blinded to the phenotype defined by RSC). In order to define B- or T-cell lymphoma, more than 75% of the neoplastic lymphocytes needed to be positive for the B-cell marker (PAX5) or the T-cell marker (CD3) [12,23], respectively, with residual cells (non-neoplastic) showing positivity to the other marker. A non-B non-T lymphoma was assigned when neoplastic lymphocytes were negative to both markers, but positivity was detected in residual normal lymphocytes. A mixed population was identified when around half of the neoplastic cells were CD3-positive and the remaining were PAX5-positive. The phenotype could not be defined in all the other scenarios and an inconclusive result was considered.

For assessing interobserver reproducibility in defining the lymphoma phenotype in RSC and CB, a board-certified veterinary pathologist with a long experience in this specific topic of canine and feline lymphomas (MC—considered as observer 2) independently evaluated the results of the immunolabeling on RSC and CB and determined the percentage of positive cells and their intensity.

### 2.4. Statistical Analysis

Statistical analysis was performed using R version 4.1.2 [24]. The kappa value and percentage of agreement in scoring positivity, intensity, background, and non-specific staining in immunolabeling on RSC and CB were assessed using packages vcd and irr, respectively [25,26]. The Wilcoxon signed-rank test was used to evaluate the differences in the scores/levels of each parameter in RSC and CB. The interobserver variability in positivity and phenotype diagnosis were also tested using kappa statistics. Kappa values were interpreted as previously described [27]. Briefly, an adequate agreement was considered only when kappa values were ≥0.61. The number of cases with inconclusive/non-defined phenotype by immunolabeling on RSC and CB was compared with the Fisher test. Finally, univariable and multivariable logistic regression analysis was used to evaluate the influence of pre-analytical variables, such as species, type of specimens (needle rinses and effusion fluids), coverslipping, and time of archive on the risk of an inconclusive phenotype on RSC. For this purpose, the phenotypes diagnosed by the board-certified veterinary pathologist (MC) (observer 2) were considered. A *p* < 0.20 was considered to select variables from univariable analyses to multivariable analysis. A *p* < 0.05 was considered to define statistical significance in the final multivariable models and all the other statistical tests.

## 3. Results

### 3.1. Study Case Series

The 57 cases (30 dogs and 27 cats) were classified by cytology as nodal lymphomas (n = 28), lymphoma effusions (n = 22), cutaneous (n = 1), splenic (n = 1), and intestinal lymphomas (n = 1), and lymphocyte-rich/chylous effusions (n = 4) (Table 1). Data regarding the pre-analytical variables’ time of archive and coverslipping are displayed in Table 2. The raw data of the pre-analytical variables (type of specimens, time of archive time and coverslipping) of all the included cases are displayed in Appendix A.

During the immunolabeling on the RSC procedure, the material was lost in 7 cases (from all the smears or one of the staining areas), thus compromising the evaluation of one of the markers.

### 3.2. Comparison of Immunolabeling on RSC and CB for CD3

The kappa value of the agreement on scoring the percentage of CD3-positive cells in RSC and CB was 0.20 (95% CI, confidence interval 0.06–0.34, *p* = 0.003) and 0.28 (95% CI 0.10–0.46, *p* = 0.001) for observer 1 and observer 2, respectively. These figures represented a fair agreement for both observers, and overall, the positivity in RSC and CB was significantly different (*p* < 0.0001). In half of the cases, the percentage of CD3-positive cells by RSC was lower than CB (Appendix A). For instance, 20 cases with positive CD3 cells in CB (8 out 20 with score 4, >75% positive cells) were utterly negative in RSC. When the positivity was analyzed separately for each species, no significant agreement existed between the CD3 positivity scores in RSC and CB in cats, while the agreement in dogs remained fair [kappa = 0.33 (95% CI 0.06–0.60, *p* = 0.01) for RSC and CB and kappa = 0.44 (95% CI 0.14–0.73, *p* = 0.003) for observer 1 and observer 2, respectively] (Table 3). When analyzing the pre-analytical parameters, a fair agreement in the score of CD3 positivity in CB and RSC was also observed for cases with cytology samples coverslipped or not (Table 3). Moderate agreement was obtained in the comparison of the CD3 positivity scores in FNA cytology specimens and the corresponding needle rinses CB as evaluated by the board-certified pathologist [kappa = 0.47 (95% CI 0.19–0.75), *p* = 0.001] (Table 3). When the effusion cases were analyzed separately, there was no significant agreement on scoring the number of CD3-positive cells by RSC and CB for both observers. Similarly, there was no agreement between the immunolabeling on RSC and CB CD3 positivity scores when the analysis considered the cases divided by each archival time interval (even in recent cases, ≤5 months, there was no significant agreement).

Overall, the CD3 intensity signal was significantly lower in RSC than in CB. Only seven cases presented strong intensity (level 3) in RSC, whilst 33 cases were intensely stained for CD3 in CB (Appendix A). No significant agreement existed for this parameter between immunolabeling on RSC and CB.

The presence of background and non-specific staining in CD3 RSC was not concordant with that of CB (Table 3). In fact, these parameters were significantly different in immunolabeling on RSC and CB (*p* = 0.003 and 0.002 for background and non-specific staining, respectively). Background staining was observed in 22 RSC (seven effusions and 15 FNAs), including canine and feline cases, as well as the most recent cases (with less than 15 days of archive). In these, the background was weak in 13 cases, moderate in 8 cases, and strong in one case (Appendix A). Notably, background staining was observed/detected in only 11 anti-CD3 CB (five effusions and six needle rinses CB, being weak in all cases). Regarding the non-specific staining, this parameter was detected in 18 RSC (nine effusions and nine FNAs) from dogs and cats, with different archive times (Appendix A). Non-specific staining was observed in 6 out of 57 CB for the CD3 CB (Appendix A).

### 3.3. Comparison of Immunolabeling on RSC and CB for PAX5

The agreement between the scores of the percentage of PAX5-positive cells in immunolabeling on RSC and CB was substantial for both observers [kappa = 0.64 (95% CI 0.51–0.78) and kappa = 0.62 (95% CI 0.45–0.78), *p* < 0.0001 for observer 1 and observer 2, respectively)] (Appendix A). In Table 4 the variation of the kappa values according to the pre-analytical variables is represented. The agreement decreased in cases with 5–12 months or more than 24 months of archive (Table 4). The animal species influenced the agreement, since a substantial agreement between the number of PAX5-positive cells in RSC and CB [kappa values 0.70 (95% CI 0.54–0.86) and 0.74 (95% CI 0.58–0.91), *p* < 0.0001, for observer 1 and 2, respectively] was observed only for canine cases. Evaluation of immunolabeling on RSC and CB for PAX5 in feline samples returned no agreement for observer 2 and fair agreement for observer 1. (Table 4). Similarly, when the positivity scores assigned to PAX5 staining on RSC of effusion fluids were compared to those assigned for their corresponding CB, the agreement was poor or fair, depending on the observer (Table 4).

Overall, the PAX5 immunostaining intensity level was significantly lower in immunolabeling on RSC compared to CB (*p* < 0.0001). Intense nuclear PAX5 staining was observed in lymphocytes in 40 cases in CB. Of these, 11 cases also presented intense signals in RSC (18, 6, and 5 presented moderate, low, and negative PAX5 staining, respectively) (Appendix A). The agreement was poor for PAX5 intensity level and background and non-specific staining in RSC and CB (Appendix A). Despite not being statistically significant, background staining tended to be more frequent in RSC than in CB sections (19 versus 12 cases, respectively) (Appendix A).

### 3.4. Immunophenotyping and Interobserver Agreement

The interobserver agreement on scoring positivity was substantial for CD3, but only in immunolabeling on CB [k = 0.65 (95% CI 0.55–0.76), *p* < 0.0001]. The agreement in assigning a PAX5 positivity score in immunolabeling on RSC and CB was substantial [k = 0.86 (95% CI 0.77–0.94), *p* < 0.0001 and k = 0.79 (95% CI 0.69–0.90), *p* < 0.0001, respectively]. As the ultimate goal of immunolabeling on RSC and CB is to define the lymphoma phenotype, we also analyzed their interobserver reproducibility. For this comparison, only the lymphoma cases were included (i.e., 53 cases: 23 feline, and 30 canine). The number of inconclusive immunophenotypes tended to be higher in RSC (26 out of 53 and 29 out of 53 for observer 1 and observer 2, respectively) (*p* = 0.05 and *p* = 0.02 for observers 1 and 2, respectively). The distribution of the inconclusive cases in immunolabeling on RSC according to the pre-analytical variables is represented in Table 5.

In immunolabeling on CB, only 7 out of 53 and 10 out of 53 cases (observers 1 and 2, respectively) led to an inconclusive phenotype. Of the 10 inconclusive cases in CB identified by observer 2 (board-certified pathologist), 9 also presented inconclusive results in RSC and one was a B-cell lymphoma. These comprised 6 needle rinses and 4 effusion CB, belonging to 4 cats and 6 dogs.

Notably, there was a significant interobserver agreement in defining a conclusive or inconclusive phenotype, both in RSC and in CB, being relatively higher in RSC [observers were concordant in 85% of the cases, kappa = 0.74 (95% CI 0.55–0.92), *p* < 0.0001 in RSC and kappa = 0.65 (95% CI 0.37–0.93), *p* < 0.0001 in CB].

For the more experienced board-certified pathologist, the majority of the B-lymphomas (88%) diagnosed after immunolabeling on CB was also identified with RSC (Figure 1). All the canine B-cell lymphomas identified by immunolabeling on CB (n = 14) were also diagnosed by RSC, but only 1 of the 3 B-cell lymphomas in cats was identified by RSC. As to T-cell lymphomas, a different scenario appeared. Only 7 out of the 22 T-cell lymphomas classified by CB were also diagnosed as such in RSC. The remaining 15 cases in RSC rendered inconclusive results (Figure 2 and Table 6).

As to the non-B non-T lymphomas identified by CB, these were classified as inconclusive, mixed population and one case was diagnosed as B-cell lymphoma in RSC.

The contingency table (Table 7) comparing the immunophenotypes defined by each observer in RSC and CB showed that the interobserver agreement on the diagnosis of a B lymphoma was perfect or almost perfect both in RSC (kappa = 1, *p* < 0.0001) and in CB [kappa = 0.95 (95% CI 0.87–1), *p* < 0.0001]. For the diagnosis of a T cell population, the agreement between the two observers was also high, being higher in CB [kappa = 0.92 (95% CI 0.82–1), *p* < 0.0001] comparing to RSC [kappa = 0.73 (95% CI 0.45–1), *p* < 0.0001]. Overall, the interobserver reproducibility of immunophenotyping (considering all the categories: inconclusive, B lymphoma, T lymphoma, non-B non-T lymphoma, and mixed) was higher on CB [kappa = 0.70 (95% CI 0.49–0.92), *p* < 0.0001] compared to RSC [kappa = 0.60 (95% CI 0.36–0.83), *p* < 0.0001].

### 3.5. Role of Pre-Analytical Variables in Immunophenotyping

The pre-analytical characteristics of cases with an inconclusive and conclusive lymphoid phenotype diagnosis by immunolabeling on RSC are represented in Table 5. The risk for a non-diagnostic result was statistically significantly lower in dogs compared to cats [OR = 0.66 (95% CI 0.51–0.85), *p* = 0.002] and significantly higher in effusion fluids compared to FNA specimens [OR = 1.59 (95% CI 1.24–2.03), *p*= 0.0005]. The time of archive also affected the classification of the immunophenotype, being this influence statistically significant only for cases with long archival time (more than 5 years) [OR = 1.82 (95% CI 1.08–3.07); *p* = 0.03].

After the univariate analysis, the variables species, specimen type and time of archive were included in a multivariable model. The model tended to lower the odds of an inconclusive result in canine samples [OR = 0.70 (95% CI 0.48–1.0), *p* = 0.07]. To disclose interactions between the pre-analytical variables (that showed possible roles in the univariable analysis), different multivariable models were constructed. The inclusion in the multivariable model of the interaction between the time of archive and species showed that cases with more than 5 years of archive compared to recent ones (less than 15 days of archive) tended to have a higher risk for an inconclusive immunophenotype. Moreover, this model showed that the two variables, species and time of archive, probably interact with each other on the risk for an inconclusive immunophenotype result in RSC.

## 4. Discussion

Lymphoma is frequent in dogs and cats with a heterogeneous clinical and morphological presentation [28]. Cytology has a major role in diagnosing lymphoma in veterinary medicine, mostly from FNA of lymph nodes, organs, or from effusions [2,29,30]. In cats, cytological diagnosis is more challenging, excluding effusion and mediastinal lymphomas, and the immunophenotype can be essential for confirming or excluding the diagnosis [31]. The distinction of most canine lymphoma cases from reactive and inflammatory conditions is relatively straightforward to achieve by FNA of lymph nodes alone [32]. However, their characterization and immunophenotype prediction can be puzzling, even in dogs. Recent studies reported that experienced board-certified pathologists were deceived by morphological features when predicting the immunophenotype by cytology, both in dogs and cats [11,30]. Therefore, complementary phenotyping methods are needed to improve the accuracy of cytology to discriminate between B- and T-lymphomas. Nowadays, B- and T-cell lymphoma can be differentiated through immunohistochemistry in tissues, flow cytometry techniques, and PARR.

To maximize the use of minimally invasive clinical samples, immunophenotyping should be attempted in FNA. On many occasions, these will probably be the only samples available to pursue with further tests. Using fresh, unstained samples is often impossible, as there are not enough slides available. It may even compromise the diagnosis if poor-quality slides (misjudged as adequate before staining) are used in the immunostaining [10,33]. Nowadays, there is some evidence that RSC can be used for assessing different immunomarkers [33], including lymphoid markers, at least in dogs [10]. In cats, limited information exists [18]. At least in theory, RSC could be a readily available source of material to conduct retrospective studies, investigate, or address the immunophenotype in an individual clinical case. In the present study, we compared the performance of RSC and matched CB for immunophenotyping in dogs and cats.

The performance of immunophenotyping on RSC was unsatisfactory, with inconclusive results in around half of the cases. This high number of inconclusive results seems unrelated to the observer, but remains an intrinsic factor of the sample resulting from a loss of antigenicity of the cells in cytology samples. A previous study has shown that the immunoreactivity for anti-CD3 and CD-20 in RSC from dogs was kept for 5 months [10]. Surprisingly, herein non-diagnostic immunolabeling on RSC results occurred, not only in cases with a long time of archive, but also in recent ones (less than 5 months), especially in cats.

The immunophenotype of the lymphoma was established by CB in 81% of cases. In a previous study, Heinrich et al. [9] immunophenotyped a similar percentage of canine lymphoma cases (22 out of 23 B-cell lymphomas and 1 out of 6 T-cell lymphomas) using CB and anti-CD20 and anti-CD3 antibodies. In our study, 9 of the inconclusive cases in immunolabeling on CB also rendered a non-diagnostic result in immunolabeling applied to RSC, with a single case classified as a B-cell lymphoma by RSC. This discrepancy can be related to the loss of antigens in more atypical lymphomas, tumor cell heterogeneity, or intrinsic variables of the CB. It should be recalled that using a single B-cell marker in cats is associated with a failure of B-cell lymphoma diagnosis in around 15% of the cases, even in histopathology [34]. Other reports also showed that using of a single B-cell and T-cell marker resulted in 10 to 20% unclassified canine lymphoma immunophenotypes in histologic samples [23].

Overall, our findings showed that in most cases with inconclusive immunophenotype in RSC, the CB allowed the evaluation of the B- or T-cell phenotype. Still, inconclusive results can be obtained in some CBs when only two lymphoid markers are used. In these cases, a large panel of T- and B-cell markers should be performed [28]. For a large panel of markers, consecutive sections of CB are the most suitable type of platform [19]. This larger panel could include CD3, PAX5, CD20, CD79α, and CD5, as it has been used recently in canine and feline studies [11,30].

The nuclear lymphoid marker PAX5 presented comparable results on RSC and CB in dogs, despite presenting a significantly less intense signal in the former samples. Apparently, this lower intensity in RSC did not jeopardize the B-cell phenotype assessment in canine samples, since all the B-cell lymphomas identified in RSC were also identified in CB. This is in accordance with a previous report of immunolabeling in RSC [10]. Likewise, a good performance of PAX5 immunostaining on archived canine cytological smears has also been reported in human medicine [35]. Contrasting with our findings, Cozzolino et al. [35] suggested that the intensity could be higher in previously Papanicolaou stained smears compared to CB since the genetic material of the cells is preserved as a whole in cytology smears [35]. In feline lymphoma cases, the performance of the PAX5 antibody on RSC was unsatisfactory. In this species, PAX5 immunostaining in RSC was not related to the CB, with only a single B-cell lymphoma (out of 3) clearly diagnosed by RSC.

The immunodetection of CD3 on RSC was highly hampered, particularly in feline and effusion cases. Notably, in cases with CD3-positive cells in CB, more than one third was completely negative in the immunolabeling applied to RSC. Considering that CD3 positivity in the majority of cells is essential for the immunophenotype assessment, it is not surprising that only 7 out of the 22 T-cell lymphomas classified by immunolabeling on CB were likewise diagnosed as such by RSC. In cytology slides, CD3 immunolabeling was also characterized by a lower intensity signal, higher background, and non-specific staining. We could not exclude that the combination of higher background and very low immunoreaction signal could have synergistically contributed to considering CD3 positive cases as being negative, thus decreasing the diagnostic yield of T-cell lymphocyte populations by RSC. This lower intensity and specificity of the CD3 marker in RSC has already been reported elsewhere [10,18]. The high percentage of entirely negative cases in our series including archived and very recent cases (cases diagnosed within a week before the immunolabeling on RSC) was a concerning result. The loss of CD3 antigenicity affected both coverslipped and non-coverslipped cytology smears. In the study by Dorfelt et al. [18], the influence of destaining, coverslipping, and antigen retrieval on CD3 immunolabeling was assessed in feline samples. Our data, did not confirm that the exposure of non-covered Romanowsky-stained lymphocytes to air was a key factor in abolishing CD3 antigenicity [18]. Actually, the present study showed that lymphoid antigen preservation in cytology specimens could be affected by different interacting pre-analytical factors. Indeed, regression analyses demonstrated that the performance of immunolabeling on RSC with CD3 and PAX5 depends on different pre-analytical variables, namely species, time of archive, and specimen type. According to our results, RSC from cats and effusion fluids are the less suitable for immunolabeling; in these cases, immunophenotyping should be performed in CB or surgical tissue biopsies. Additional caveats of using archived cytology smears for immunolabeling noticed in the present study include the possibility of cell loss during the procedure (which occurred in 14% of the cases in our study); and the higher odds for background and non-specific staining. These two features can result from disrupting the cells in cytology smears, leaking of antigens (namely those membranous or cytoplasmic), and dense areas on the smears. All these features may challenge the interpretation of the results of the immunoreaction [3,36,37,38].

The interobserver agreement in canine and feline lymphomas characterization by routine cytology, including predicting the phenotype, has been reported to be fair [11,30]. Herein, the interobserver reproducibility in scoring the number of immunolabeled cells for anti-CD3 and anti-PAX5 in RSC and CB, and consequently in the assignment of a phenotype, was assessed. The observers had a substantial agreement in scoring the nuclear PAX5 marker in cytology and CB, but a similar agreement for CD3 was only obtained in CB material. Indeed, interpreting the CD3 positivity on RSC was more prone to interobserver variation. On one hand, the lower immunoreactivity and higher background staining on RSC probably accounted for this lower agreement on establishing a T-lymphocyte phenotype. On the other hand, diagnosing B-cell lymphoma in dogs, supported by compatible cytological features and nuclear expression of PAX5, could be performed in archived cytology and CB, with a low interobserver variation. Few studies report the reproducibility of evaluating immunoreactions in canine and feline cells and tissues. Still, it is noteworthy that the interobserver agreement in scoring PAX5 positivity on RSC and CB was similar to that reported for scoring endocrine cytoplasmic immunomarkers in small tissue samples (tissue microarrays) of canine insulinomas [39]. Interestingly, that study also showed that the heterogeneity of the expression of a marker in the tissue/cell of interest is the most relevant factor, contributing to a substantial difference between the scores of the raters [39]. Considering lymphoma’s homogeneous/clonal nature, the CD3 expression heterogeneity in a T-cell population is unlikely to explain the marked variation between observers in the semi-quantitative evaluation of the CD3 positivity on RSC.

## 5. Conclusions

Detecting lymphoid markers, especially for membranous antigens in retrospective RSC samples, is hampered by pre-analytical variables and old archive cases (>5 years), and effusion and feline samples should be avoided. The two observers agreed that immunophenotyping could not be accomplished on RSC in some recent feline lymphoma and archived cases. On the other hand, CB was a valuable complementary technique for immunophenotyping feline and canine lymphomas. Thus, when the gold standard histopathologic exam cannot be performed, the recommendation is that clinicians and pathologists should make all the efforts to obtain CB, which will add no significant extra-costs (few cents per CB). These allow, in the vast majority of cases, the definition of a phenotype that would confirm the diagnosis of a dubious canine or feline lymphoma case or assist in the therapeutic planning and prognostic assessment. CB is particularly relevant for clinical settings where more expensive techniques, such as PARR or flow cytometry, are not available or affordable. In veterinary cytology, similarly to the routine standards in human pathology [37], efforts should be combined to set a path towards submitting cytological smears accompanied by a CB for diagnosis, obtained from needle-rinses after FNA of lymph nodes or other organs or obtained directly from effusions rather than including only several and repeated FNA slides.

## Figures and Tables

**Figure 1 vetsci-10-00157-f001:**
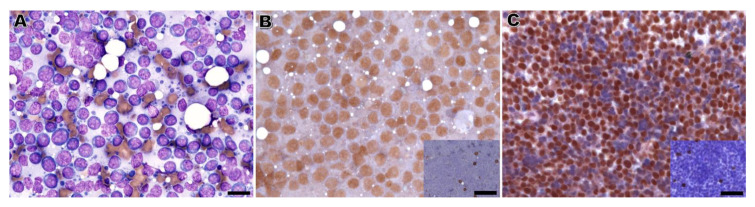
B-cell lymphoma, dog. (**A**) Fine-needle aspiration smear of an enlarged lymph node. The predominance of medium to large-sized lymphocytes with a round to slightly indented nuclei and prominent nucleoli. Hemacolor; bar = 28 µm. (**B**) Immunolabeling on Romanowsky-stained cytology slides (i.e., applied to a previously stained cytology smear). More than 80% of the neoplastic lymphocytes showed positive nuclear reactivity for PAX5, with few residual lymphocytes positive for CD3 (inset). Diaminobenzidine chromogen, Hematoxylin counterstain; bar = 28 µm (58 µm for the inset). (**C**) Immunolabeling on matched cell tube block sections. A similar percentage of positive lymphocytes for PAX5 and CD3 (inset). Diaminobenzidine chromogen, Hematoxylin counterstain; bar = 38 µm (68 µm for the inset).

**Figure 2 vetsci-10-00157-f002:**
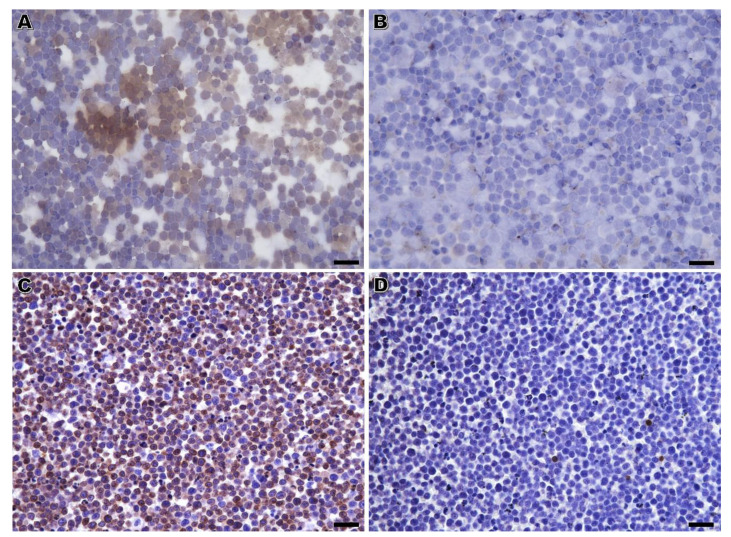
Pleural effusion in a cat compatible with lymphoma. (**A**,**B**) Immunolabeling on Romanowsky-stained cytology. None of the cells showed immunoreactivity to CD3 (**A**) or PAX5 (**B**), and background staining was observed (**A**); (**C**,**D**) Immunolabeling of CB on matched cell tube block sections. Positivity for CD3 on the cell surface was observed in more than 75% of the lymphocytes (**C**) with few being immunostained with anti-PAX5 antibody (**D**). These results on CB allowed the definition of a T-cell phenotype. Diaminobenzidine chromogen, Haematoxylin counterstain; bar = 20 µm (**A**,**B**) and 34 µm (**C**,**D**).

**Table 1 vetsci-10-00157-t001:** Type of specimen and cytology diagnosis of enrolled canine and feline cases.

	Dogs	Cats
Effusion fluids	4	22
Lymphoma	4	18
Chylous effusions	0	4
Fine-needle aspiration	26	5
Nodal lymphoma	24	4
Intestinal lymphoma	0	1
Splenic lymphoma	1	0
Cutaneous lymphoma	1	0

**Table 2 vetsci-10-00157-t002:** Distribution of cases according to the pre-analytical variables time of archive and coverslipping.

	Coverslipped	Non-Coverslipped
Time of archive		
≤15 days	4	1
>15 days ≤5 months	7	4
>5 ≤12 months	8	0
>12 ≤24 months	5	5
>24 ≤60 months	3	9
>60 months	7	4

**Table 3 vetsci-10-00157-t003:** Variation of the agreement (kappa values) of the parameters of the CD3 immunolabeling applied to previous Romanowsky-stained cytology slides and of the CD3 immunolabeling of CB on cell blocks (as evaluated by a board-certified pathologist) according to the pre-analytical variables: species, time of archive, coverslipping, and type of specimen (NS non-significant).

Pre-Analytical Variables	Positivity	Intensity	Background	Non-Specific Staining
Species				
Dog	K = 0.44 (*p* = 0.004)	NS	NS	NS
Cat	NS	NS	NS	NS
Time of archive				
≤15 days	NS	NS	K = −0.36 (*p* = 0.009)	NS
>15 days ≤5 months	NS	NS	NS	NS
>5 months ≤12 months	NS	NS	NS	NS
>12 months ≤24 months	NS	NS	NS	NS
>24 months ≤60 months	NS	NS	NS	NS
>60 months	NS	NS	NS	NS
Coverslipping				
Yes	K = 0.24 (*p* < 0.05)	NS	NS	NS
No	K = 0.30 (*p* < 0.05)	NS	NS	NS
Type of specimen				
FNA + needle rinse CB	K = 0.47 (*p* = 0.001)	NS	NS	NS
Effusion fluid + fluid CB	NS	NS	K = 0.30 (*p* < 0.05)	NS

**Table 4 vetsci-10-00157-t004:** Variation of the agreement (kappa values) of the parameters of the PAX5 immunolabeling applied to previous Romanowsky-stained cytology slides and of the PAX5 immunolabeling on cell blocks (as evaluated by a board-certified pathologist) according to the pre-analytical variables: species, time of the archive, coverslipping and type of specimen (NS non-significant).

Pre-Analytical Variables	Positivity	Intensity	Background	Non-Specific Staining
Species				
Dog	K = 0.74 (*p* < 0.0001)	NS	NS	NS
Cat	NS	NS	NS	NS
Time of archive				
≤15 days	K = 0.61 (*p* < 0.0001)	NS	K = 0.55 (*p* = 0.049)	NS
>15 days ≤5 months	K = 0.69 (*p* < 0.0001)	NS	NS	NS
>5 months ≤12 months	NS	NS	NS	NS
>12 months ≤24 months	K = 0.74 (*p* < 0.0001)	NS	NS	NS
>24 months ≤60 months	K = 0.49 (*p* = 0.009)	NS	NS	NS
>60 months	NS	NS	NS	NS
Coverslipping				
Yes	K = 0.69 (*p* < 0.0001)	NS	NS	K = −0.09 (*p* = 0.049)
No	K = 0.50 (*p* = 0.0002)	NS	K = 0.38 (*p* = 0.004)	NS
Type of specimen				
FNA + needle rinse CB	K = 0.69 (*p* < 0.0001)	NS	NS	NS
Effusion fluid + fluid CB	NS	NS	NS	NS

**Table 5 vetsci-10-00157-t005:** Pre-analytical characteristics of cases with inconclusive and conclusive immunophenotyping diagnosed by immunolabeling on 53 previous Romanowsky-stained cytology slides.

	Phenotype by Immunolabeling on Romanowsky-Stained Cytology Slides	
Pre-Analytical Variables	Inconclusive	Conclusive
Species		
Dog (n = 30)	11	19
Cat (n = 23)	18	5
Time of archive		
≤15 days (n = 5)	2	3
>15 days ≤5 months (n = 11)	5	6
>5 months ≤12 months (n = 8)	2	6
>12 months ≤24 months (n = 10)	7	3
>24 months ≤60 months (n = 11)	5	6
60 months (n = 8)	8	0
Coverslipping		
Yes (n = 30)	18	12
No (n = 23)	11	12
Type of specimen		
FNA + needle rinse CB (n = 31)	11	20
Effusion fluid + fluid CB (n = 22)	18	4

**Table 6 vetsci-10-00157-t006:** Matrix comparing the immunophenotype defined by a board-certified pathologist in immunolabeling on previous Romanowsky-stained cytology (RSC) slides and on the matched cell blocks (CB).

	RSC					
CB	B	T	Non-B-Non-T	Mixed	Inconclusive	Total
B	15	0	0	0	2	17
T	0	7	0	0	15	22
Non-B-non-T	0	0	0	0	3	3
Mixed	1	0	0	0	0	1
Inconclusive	1	0	0	0	9	10
Total	17	7	0	0	29	53

**Table 7 vetsci-10-00157-t007:** Contingency table comparing the immunophenotype defined by the two observers after evaluating CD3 and PAX5 immunolabeling on Romanowsky-stained cytology slides (left columns) and immunolabeling applied to effusion fluid and needle rinses cell blocks (right columns).

	Observer 1					
Observer 2	B	T	Non-B-Non-T	Mixed	Inconclusive	Total
B	17/17	0/0	0/0	0/0	0/0	17/17
T	0/0	5/21	0/0	0/0	2/1	7/22
Non-B-non-T	0/0	0/1	0/2	0/0	0/0	0/3
Mixed	0/0	0/0	0/0	0/1	0/0	0/1
Inconclusive	0/1	1/0	4/3	0/0	24/6	29/10
Total	17/18	6/22	4/5	0/1	26/7	53/53

## Data Availability

All data generated or analyzed during this study are included in this published article and its Appendix A.

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
