# Peer review of "Detection of Lymphoid Markers (CD3 and PAX5) for Immunophenotyping in Dogs and Cats: Comparison of Stained Cytology Slides and Matched Cell Blocks"

_vetsci, 2023, doi:10.3390/vetsci10020157_

Round 1

Reviewer 1 Report

The paper of Sampaio et al. investigated the suitability of using immunocytochemistry on archival samples with immunohistochemistry on paraffin blocks. In addition inter-observer agreement and pre-analytical variables were assessed. In general, this is a relevant article with extensive and well-performed statistics.

General comments:

The research is well described, but what is the relevance of inspecting archival material. I understand that this is the only way to get sufficient samples from the clinic, but I would assume that in a clinical setting, fresh cytology samples will be used for immunolabeling. The authors should address and discuss this. The fact that the CB slides are more optimal could be important here, as the immunogens are probably better preserved in paraffin. Maybe in reality, direct immunolabeling on cytology samples might perform just as well, or better compared to immunolabeling on cell blocks.

It would be helpful to better clarify the use of the pre-analytical variables and why their influence can be so important.

The readability of the paper is somewhat hampered by the many abbreviations that are used. Also the terminology IHC vs ICC is maybe not best. Both are in principle the same techniques, but the term ICC is more used for cytology and IHC is more used for histological sections. Could it be changed by immunolabeling, that is generic and you could say immunolabeling of cytology slides or immunolabeling of sections as an alternative?

Inter-observer agreement is discussed, but was only measured on two observers. This is a limitation that should be addressed.

Title

The title is too vague and general. “samples” should be specified (lymphomas and chylous effusion?), the antibodies used, should be defined in the title.

Abstract

The abstract could start with a short one sentence problem statement.

26: Interobserver agreement based on how many observers? What is their background (pathologist?)

33: How did the pre-analytical variables hamper the markers?

34: the term “platform” is not well chosen. Tissue blocks are not a platform.

Introduction

It should be stated how lymphomas in dogs/cats is usually diagnosed (apart from immunophenotyping)? Which samples should be taken? General information is missing.

Is immunophenotyping currently used in other veterinary cancers? Please clarify.

Why does this article focus on CD3 and PAX5 specifically?

71-79: This seems more in place in the discussion rather than in the introduction.

90: Please clarify which different methods were already used? Why is there a difference in success of immunophenotyping (line 91)?

93: Please specify: only for immunophenotyping for lymphoma or in general? Is there no literature on any feline/canine tumours that have already compared ICC and IHC?

98: For IHC on CB there is need a piece of tissue anyway?

100: Please specify the tested markers: only CD3 and PAX5?

100: No CB were obtained from the lymphomas itself? This is not clear, please clarify above.

101: How many observers? Background?

103: Why would you want to assess the pre-analytical variables? What influence do you expect? Please clarify.

M&M

108: The number of cases included/samples is not mentioned. Was there a sample-size calculation performed?

113: “ruled” instead of “rule”

115: Blinded?

156: Were all samples analyzed in one batch for ICC and IHC?

159: How was the percentage calculated? For each HPF? For a limited number of cells? Please clarify.

179: Did observer 2 also calculate the percentages and intensity?

Results

General: Do you expect breed/age/sex related differences in dogs and cats? Please clarify as nothing is mentioned about this.

200: Number of samples should go in the M&M section; histopathological results can be in the results section.

204: This should go in the M&M section as this is a pre-analytical variable.

205: The results of the pre-analytical variables can of course go in the Results section.

208: Does this means exclusion of these samples? Please clarify the correct number of cases left.

355: The role of these pre-analytical parameters remains unclear: why are they investigated? Please clarify above.

Discussion

375: Cytology is crucial but please mention the origin of cytology smears/samples for diagnosis (eg. lymph nodes etc).

380: Cytology of what?

384: How do they distinguish between B- and T lymphoma nowadays? Based on which technique(s)? please clarify.

393: Why are these samples not excluded due to bad quality and cell loss?

435: Should the CD3 staining be further optimized based on your and previous results? Is it still reliable if it has low intensity, high background a non-specific staining?

448: What would your advice be for pre-analytical factors based on these results? Please state.

465-466: Avoid repetition

Results

482: Please clarify how to minimalize the impact of these pre-analytical variables.

487: How practical is it to obtain CB? Cost?

493: Please state of which tissue/organs the CB should be obtained and which protocol should be used for this in veterinary practice.

Figures & tables

Table 5: It remains unclear why exactly these criteria were analyzed/considered. Please clarify earlier in the text.

Figure 1: the insets should be larger; they are not clear.

Table 6, 7 and 8: Make one Table instead of 3 comparing all observers

Figure 1 and 2: Both for IHC and ICC: lack of negative and/or positive controls

The Images should be larger (especially the insets) and negative/positive controls should be added. Also the insets should get their own scale bars.

Negative/positive controls should be added (maybe in the supplements)

The insets should get their own scale bars as they are not the same size as the actual pictures.

Reviewer 2 Report

Dear authors

Thank you for submitting this interesting and well-written paper.

Following my comments/suggestion.

Line 127: You talk about scanning the slide with a digital scanner, but you do not specify the magnification at which you performed the scanning and whether you used oil. Also, please specified you used only the digital slides for evaluation or if only MC used these slides for evaluation. Please add these information.

Lines 137-138: Did you include negative controls for each run? Did you stain all the slides in the same IHC/ICC batch or in different sets (to consider a possible batch effect)?

Line 140: How did you remove the coverslip? Did you use water-based glue?

Line 141: It would be helpful to specify how long the slides were left to air dry for reproducibility purposes.

Line 142: Did you section the cell pellet before proceeding with the immune staining? If yes, which thickness?

Line 142: I am not a native English speaker, but I have the impression that likewise is not the right word in the context (since likewise can also mean similarly, but not necessarily identical). I suggest the authors use something like this: CB and cytology samples were subsequently processed similarly.

Lines 143-144: ‘during 3 minutes after maximum pressure is reached, I am not sure what you mean here. Do you mean that you left the slides in the buffer for 3 minutes after pressure was reached in the pressure cooker? If yes, I suggest using something like: Antigen retrieval was archived by slide immersion in citrate buffer (pH 6.0) for 3 minutes at high pressure, using a pressure cooker.

Line 180: I am curious how you compared the ICC and IHC because the cellularity must have been quite different. Could you please add this part to the M&Ms?

Lines 248-249: Notably, the lower background was noted in CD3 IHC since it existed only in 11 cases (five effusions and six needle rinses CB, being weak in all cases). I suggest the authors rephrase this sentence: Notably, background staining was observed/detected in only 11 anti-CD3 IHC (five effusions and six needle rinses CB, being weak in all cases).

Lines 263-265: On the other hand, no agreement or a fair agreement (depending on the observer) existed between the percentage of positive PAX5 cells in ICC and IHC in feline cases (Table 4). I would rephrase this sentence for clarity. Consider something like this: Evaluation of ICC and IHC immune staining for PAX5 in feline samples returns no agreement for observer 1 and fair agreement for observer 2.

Round 2

Reviewer 1 Report

I have no further comments